# Contrast normalisation masks natural expression-related differences and artificially enhances the perceived salience of fear expressions

**Abigail L. M. Webb**\*, **Paul B. Hibbard, Rick O'Gorman**

Department of Psychology, University of Essex, Colchester, United Kingdom

\* awebbc@essex.ac.uk

**Data Availability Statement:** Data can be accessed via Figshare (DOI: 10.6084/m9.figshare.9944216. v1).

## Abstract

Fearful facial expressions tend to be more salient than other expressions. This threat bias is to some extent driven by simple low-level image properties, rather than the high-level emotion interpretation of stimuli. It might be expected therefore that different expressions will, on average, have different physical contrasts. However, studies tend to normalise stimuli for RMS contrast, potentially removing a naturally-occurring difference in salience. We assessed whether images of faces differ in both physical and apparent contrast across expressions. We measured physical RMS contrast and the Fourier amplitude spectra of 5 emotional expressions prior to contrast normalisation. We also measured expression-related differences in perceived contrast. Fear expressions have a steeper Fourier amplitude slope compared to neutral and angry expressions, and consistently significantly lower contrast compared to other faces. This effect is more pronounced at higher spatial frequencies. With the exception of stimuli containing only low spatial frequencies, fear expressions appeared higher in contrast than a physically matched reference. These findings suggest that contrast normalisation artificially boosts the perceived salience of fear expressions; an effect that may account for perceptual biases observed for spatially filtered fear expressions.

## Introduction

Fearful facial expressions are especially salient to the human visual system relative to other expressions [1–2]. Expressions of fear capture and orient visual spatial attention [3–5], receive preferential allocation of attentional resources [6–8, 3], and emerge faster under conditions of visual suppression [9–10]. This bias for fearful expressions also occurs in peripheral vision [11–12], and when observers report being unaware of having been presented with a face [13–15].

Evolutionary accounts of prioritised processing of fearful expressions claim that it is a hard-wired, adaptive behaviour enabling rapid detection of and orientation towards cues that signal

**Funding:** AW received a PhD scholarship (ES/J500045/1) from the Economic and Social Research Council (ESRC). https://esrc.ukri.org/ The funding body did not play a role in study design, data collection, analysis, decision to publish, or preparation of the manuscript.

**Competing interests:** The authors have declared that no competing interests exist.

potential threat in the environment [1, 7, 12, 14–16]. On the other hand, low-level visual accounts propose that perceptual biases for fear expressions are driven by simple image properties and attributes that are known to influence image salience, including their contrast and spatial frequency content [9, 17, 18]. In other words, the prioritisation of fearful expressions is driven by the way in which they are processed early in the visual system, rather than as a consequence of the way in which they are evaluated and interpreted as meaningful, socially relevant stimuli [7, 9, 17, 18]. The direction of causality of evolutionary pressure is opposite under these two accounts. In the first, it is proposed that aspects of the visual system adapt to become especially sensitive to fearful faces. In the second, it is the appearance of fearful faces that adapts to become more readily detectable by the visual system.

Psychophysical studies have shown that the coarse, low spatial frequency content of fearful expressions is responsible for the fear expression bias. This crude information is sufficient for accurate detection of a face, but alone does not allow recognition of identity or age [19–21]. It does however play a particularly important role in determining the salience of fearful expressions [15, 17, 22]. Subcortical regions such as the amygdala respond preferentially to coarse information, showing strong responses to low-frequency-filtered fear expressions, but not to high-frequency-filtered versions of the same faces, or other expressions [22]. Vuilleumier and colleagues [22] emphasise that these rapid amygdala responses occurred 30ms prior to responses from the face-sensitive visual cortex, and thus operate via distinct subcortical routes. Low frequency fearful expressions also elicit faster reflexive eye movements (saccades) than happy and neutral expressions, but not when they are filtered to contain only high frequency information [17]. Hedger and colleagues [18] suggest that the perceptual advantages associated with fear expressions occur due to the particular spatial configuration of these faces, when considered simply as low-level visual stimuli. This configuration–widened eyes and raised eyebrows- produces a physical stimulus that is well-matched to the contrast sensitivity function at typical viewing distances. They showed that the effective contrast of stimuli–once contrast sensitivity is taken into account—was a reliable predictor of their detection during a backward masking task. These findings were interpreted as evidence of a sensory advantage for fearful expressions. That is, fearful expressions are salient because they are better matched to the sensitivity of the visual system, without requiring specialised mechanisms tuned specifically to their emotional content.

If fearful facial expressions are more salient as a result of their low-level luminance properties, it should be possible to quantify this using relevant metrics such as image contrast. However, in studies of the threat bias, any naturally occurring differences in physical contrast across expressions will often be removed by stimulus standardisation techniques such as contrast normalisation. Typically, physical RMS contrast is equalised across stimuli in experiments, since it is known to affect the detectability of stimuli. However, it is crucial to understand any such *naturally-occurring* differences between facial expressions for two reasons. Firstly, it is important for our theoretical understanding of facial expression perception to use stimuli as they appear in the natural world, rather than following image manipulation that is known to affect their salience. Secondly, it is important to understand how such low-level differences are influenced by stimulus standardisation techniques, and the subsequent effect this has upon perceptual responses, in order to interpret claims of a threat bias. If variations in the contrast of faces across expressions are altered by the process of normalisation, then any psychophysical, behavioural or physiological findings may reflect experimental artefacts as opposed to meaningful ecological phenomena. Indeed, Menzel and colleagues [23] show such systematic differences in contrast between expressions, whereby fearful faces are in fact *lower* in physical contrast than angry expressions. The same study also showed that normalising images of faces in terms of their luminance distribution and amplitude spectrum

impaired observers' performance in an expression matching task. These findings are important for our current understanding of face expression perception, and indeed, the fear expression bias. Firstly, they provide evidence of naturally-occurring differences in the physical composition of different facial expressions. Secondly, they show that normalisation techniques which remove these differences can influence how observers respond to different facial expressions. Naturally occurring expression-related differences in contrast may therefore be key for successful emotion identification, consistent with evolutionary theories that posit unique adaptive functions for different expressions. Moreover, findings have also shown a faciliatory effect of contrast normalisation on the effective contrast of faces [24], a quality which is known to affect their detectability [18]. If fearful expressions tend to have lower RMS contrast, as findings from Menzel and colleagues [23] suggest, then contrast normalisation will tend to disproportionately boost their physical contrast. If this is the case, then the threat bias would reflect the artificial boosting of the contrast of fearful expressions through contrast normalisation, rather than an ecologically meaningful effect.

The objective of the present study is to extend our understanding of the mechanisms that drive biases for fearful face expressions, in order to establish the importance of image contrast in the threat bias for fearful faces. The first question that we addressed was whether there are naturally occurring differences in RMS contrast across expressions, prior to contrast normalisation, which could affect the salience of these stimuli. Our first experiment therefore quantifies naturally occurring differences in physical contrast across expressions, and how this differs across different spatial scales. The second question that we addressed was whether there were would be differences in the apparent contrast of expressions. It has been suggested that fearful expressions might be more salient because their Fourier amplitude spectra well matched to the contrast sensitivity function of the human visual system [18]. If so, then this would be expected to result in an increase in their apparent contrast, which in turn might be the causal explanation of the threat bias [25, 26]. Our second experiment therefore measures expression-related differences in apparent, perceived contrast between facial expressions. Together, these results allow us to quantify the extent to which the threat bias can be explained in terms of naturally occurring differences in physical and effective contrast.

## Methods

### Experiment 1: Image analysis

**Stimuli and apparatus.**   Stimuli were greyscale front-view face pictures of 140 actors (70 male, 70 female) extracted from the Karolinska Directed Emotional Faces (KDEF) set [27]. This included neutral, angry, fearful, happy and disgusted expressions. Faces were cropped to include internal features only. Their dimensions were 300 (height) x 230 (width) pixels. We assumed a viewing distance of 65cm, such that the stimulus face-width was 7.09˚. Filtering the images using a second-order Butterworth filter in MATLAB created low (LSF), mid-range (MSF) and high spatial frequency (HSF) versions of all stimuli, in addition to the original broad spatial frequency (BSF) versions. Frequency cut-offs were $f < 1_{cpd}$ for LSF faces, $1 < f < 6_{cpd}$ for MSF faces, and $f > 6_{cpd}$ for HSF faces. Stimuli were not normalised for contrast, to ensure that naturally occurring variations in physical contrast were preserved [23]. Image analyses, including measures of RMS contrast and Fourier amplitude spectra, were performed in MATLAB. Image analysis involved the following steps:

1. Each image was cropped to a 300x200 region including the internal features of the face.

2. Pixel intensity values were scaled so that the minimum value (black) was zero and the maximum possible value was 1.

3. RMS contrast was calculated as standard deviation of the scaled pixels intensities.

4. Fourier transform was calculated using the MATLAB fft2() function, and the amplitude of each pixel in the transformation using the MATLAB abs() function.

5. For each image, pixels were binned according to their spatial frequency (across all orientations) using 100 bins, and taking the mean value of all pixels in each bin. The mean across all images for each expressions was calculated, and plotted in Fig 3.

6. For each image, the Fourier amplitude slope was estimated by performing a linear regression of log(spatial frequency) against log (amplitude).

## Experiment 2: Contrast matching

**Participants.** Nineteen individuals took part in the study (13 women), aged between 18 and 26 years. All participated as part of a credited psychology research module. Participants were instructed to use corrective lenses if they would normally do so for a computer task at a distance of 96 cm. The number of participants was determined by previous studies of a similar nature [9, 25]. The University of Essex University Ethics Committee approved the study and all participants gave written, informed consent.

**Stimuli and apparatus.** Stimuli were greyscale and front-view face pictures of 16 individuals (8 male, 8 female) extracted from the KDEF set [27]. Faces were cropped to include internal features only, and portrayed a fear, anger, happy, disgust or neutral expression. Eighty (16 identities x 5 emotions) face pictures were composed of intact broad spatial frequencies. A second-order Butterworth filter created LSF, MSF and HSF versions of faces. The cut-off frequencies matched those used in Experiment 1. Faces were presented in their normal form (upright with retained luminance polarity), and in a manipulated form (rotated by 180° with inverted luminance polarity). Inversion and luminance polarity reversal disrupts facial recognition beyond that achieved by inversion alone, while preserving low-level image properties including contrast and Fourier amplitude spectrum [9]. Stimuli were presented using a VIEWPIXX 3D monitor, viewed from a distance of 96cm. Participants used a chin rest to maintain this viewing distance. The monitor screen was 52cm wide and 29cm tall, with a 1920x1080 pixel resolution, and a refresh rate of 120Hz. The average luminance was 50 cdm$^{-2}$. Each pixel subtended 1.0 arc min. Stimuli were presented at 10-bit resolution. Stimuli were generated and presented using MATLAB and the Psychophysics Toolbox extensions [28–30].

**Procedure.** Each trial began with a target and reference face presented side by side in the centre of the screen. Reference faces always depicted an upright neutral expression, and were assigned a fixed contrast corresponding to 10% Michelson contrast. Target faces depicted one of 5 expressions (neutral, fear, anger, happy, disgust) at an upright or manipulated (inverted and reversed luminance polarity) orientation. Each target face started with a randomly assigned Michelson contrast value between 10 and 20%. Using left and right arrow keys, observers' task was to adjust the contrast of the target face–indicated by a red marker- until it appeared to have the same contrast as the reference face. An example is shown in Fig 1.

In total, 38 participants took part in the study. However, the study was separated into 2 sessions for data collection. The first collected data from 19 participants in a single block of trials for BSF, LSF and HSF faces. A second session of data collection was added to include an additional 19 participants, in order to include a measure for MSF faces not initially included in the original stage of data collection. For this second session of data collection, only MSF faces were presented to participants, and as a single block of trials. In all blocks, trials for all facial expressions including normal and manipulated versions were intermixed.

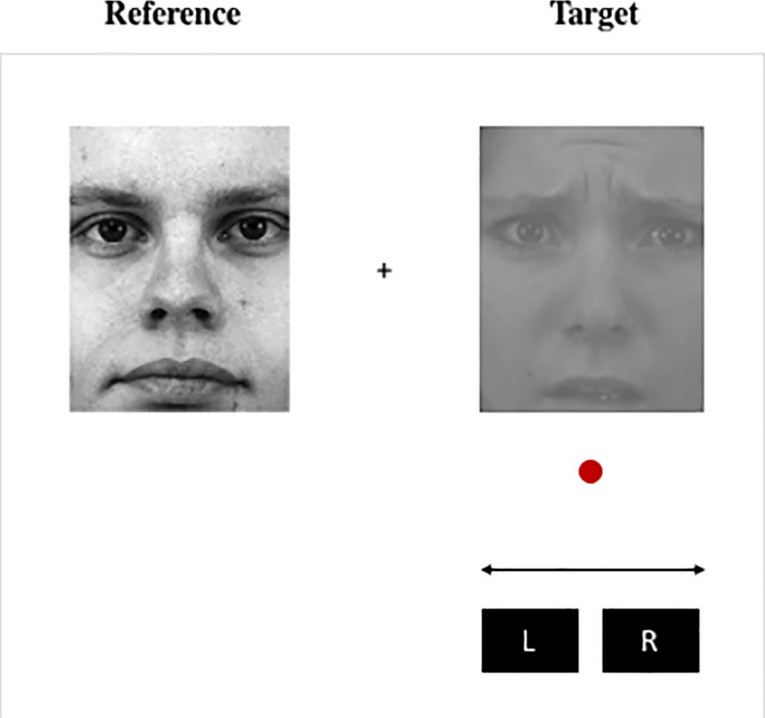

**Fig 1. Task schematic.** A task schematic depicting an example trial in which an observer is required to increase or decrease the physical contrast of a target fearful face until it perceptually matches that of a reference face.

Data from Experiment 1, 2 and post-hoc analyses for Experiment 3 are publicly-available (10.6084/m9.figshare.9944216).

## Results

### Experiment 1: RMS contrast for broadband and spatially filtered expressions

The variation in mean RMS contrast across facial expressions, and spatially filtered versions, is plotted in Fig 2. Repeated measures Analysis of Variance (ANOVA) was performed for each spatial frequency. Sidak-corrected pairwise comparisons explored differences between fear expressions and neutral, anger, happy and disgust face counterparts.

**Broadband faces.** RMS contrast varied significantly across facial expressions ($F(4, 556) =$ 11.25, $p < .001$, $\eta p^2$ .07). Sidak-corrected pairwise comparisons (summarised in S1 Table) explored RMS contrast differences between fear and expression counterparts. Statistically significant comparisons reveal that broadband fear expressions are 3.37, 5.92, and 4.73% *lower* in RMS contrast compared to neutral, angry and disgust faces (respectively), and do not differ compared to happy faces.

To understand how these differences in contrast varied according to spatial frequency, the Fourier amplitude spectrum was also measured for all 140 broadband faces (Fig 3). A repeated measures ANOVA showed a significant effect of expression on Fourier amplitude slope ($F(4, 556) = 22.63$, $p < .001$, $\eta p^2$ .14). Sidak-corrected pairwise comparisons revealed differences in slope between fear and other expressions. Fearful expressions had steeper slopes than neutral and angry faces. No other significant differences were observed. Data are illustrated in Fig 3, and summarised in S2 Table. Relative to neutral and angry faces, fearful faces have reduced

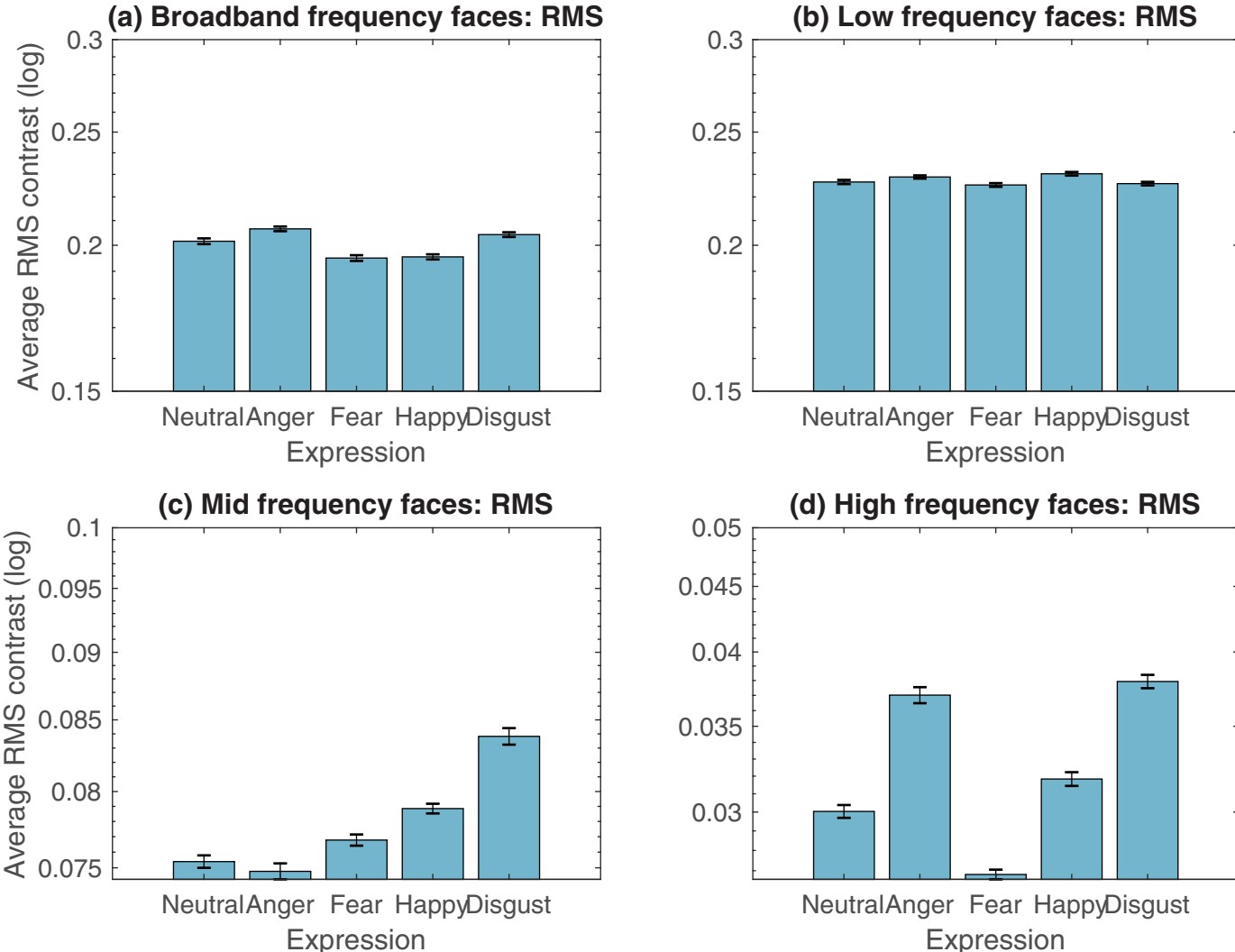

**Fig 2. Average RMS contrast for broadband facial expressions, and spatial filtered versions.** RMS contrast for broadband facial expressions, and when these faces were filtered to contain low-, mid-, or high spatial frequency components. This refers to naturally-occurring contrast differences; faces images were not normalised for contrast in any way. Note that axes are adjusted for mid-range and high frequency faces, due to substantial contrast reduction at higher frequency ranges. Error bars represent associated SEMs.

luminance contrast at high spatial frequencies. This means that the reduction in RMS contrast found for fearful faces reflects a reduction at high frequencies only. Analysis of RMS contrast for spatial-frequency filtered faces reported below was performed to explore this in detail. These analyses are summarised in S1 Table.

**Low frequency faces.** A repeated measures ANOVA revealed a significant effect of expression on RMS contrast ($F(4, 556) = 2.78$, $p = .02$, $\eta p^2$ .02). Sidak-corrected comparisons showed that low-frequency-filtered fear expressions are lower in RMS than happy counterparts only. Specifically, low frequency fear expressions are 2.23% *lower* in RMS contrast compared to low frequency happy faces.

**Mid-range frequency faces.** A repeated measures ANOVA showed a significant effect of expression on RMS contrast ($F(4, 556) = 22.86$, $p < .001$, $\eta p^2$ .14). Sidak-corrected comparisons showed that mid-range frequency fear expressions are significantly lower in RMS contrast

## Fourier Amplitude: 140 KDEF faces

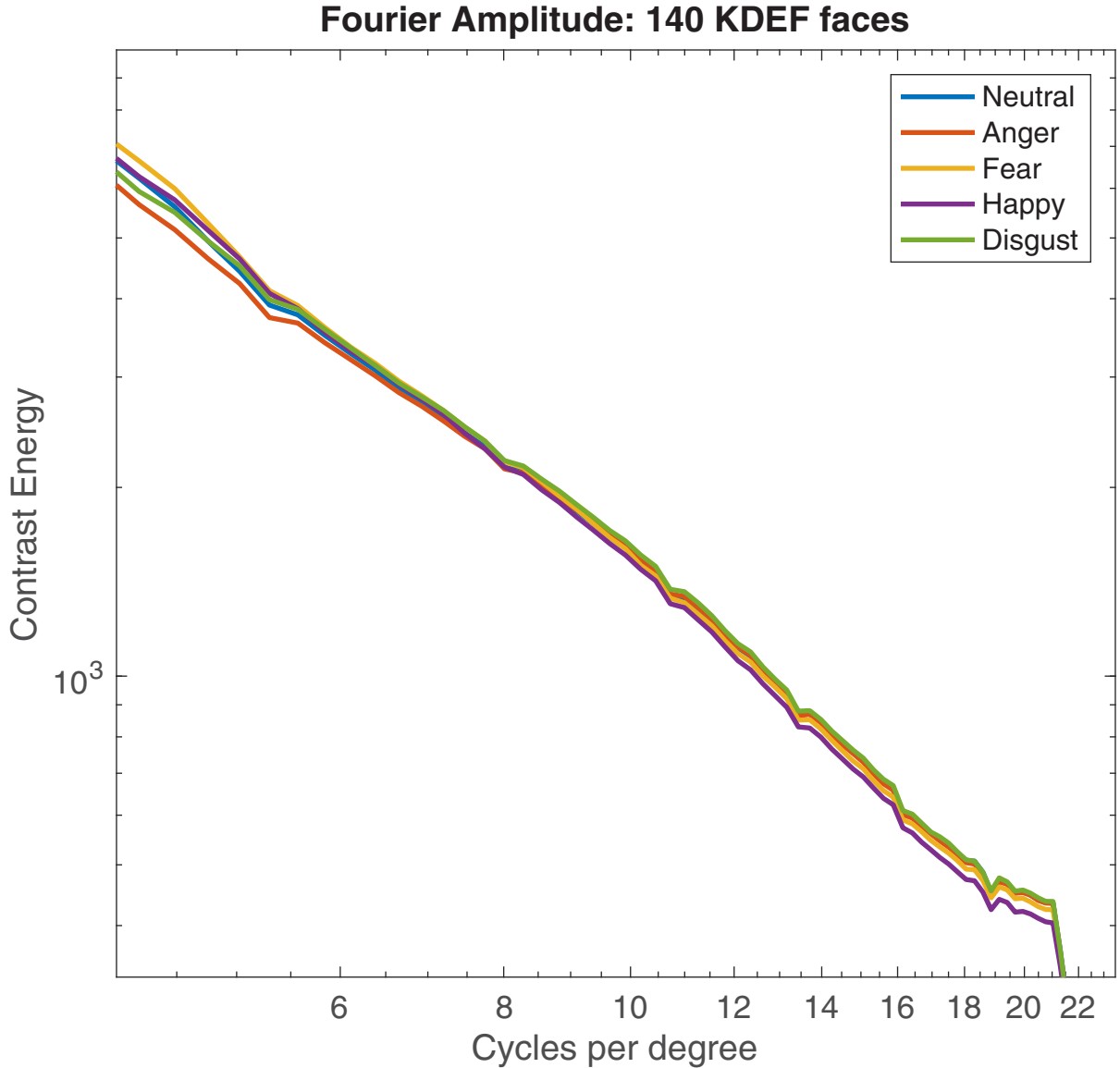

**Fig 3. Fourier amplitude spectra for 140 KDEF faces.** Average Fourier amplitude slopes for 140 facial expressions taken from the KDEF database. Faces are not spatially filtered or normalised for contrast. A steeper slope suggests that less information is contained at higher spatial frequencies.

than happy and disgusted faces. Specifically, mid-range frequency fear expressions are 2.69 and 9.2% *lower* in RMS contrast compared to mid-range happy and disgust faces (respectively).

**High frequency faces.** A repeated measures ANOVA showed a significant effect of expression on RMS contrast ($F(4, 556) = 41.93$, $p < .001$, $\eta p^2$ .23). Sidak-corrected comparisons showed that high-frequency-filtered fear expressions are significantly lower in RMS contrast than *all* other expressions. Specifically, high frequency fear expressions are 12, 38, 18.71, and 41.44% *lower* in RMS contrast compared to high frequency neutral, angry, happy and disgust faces (respectively).

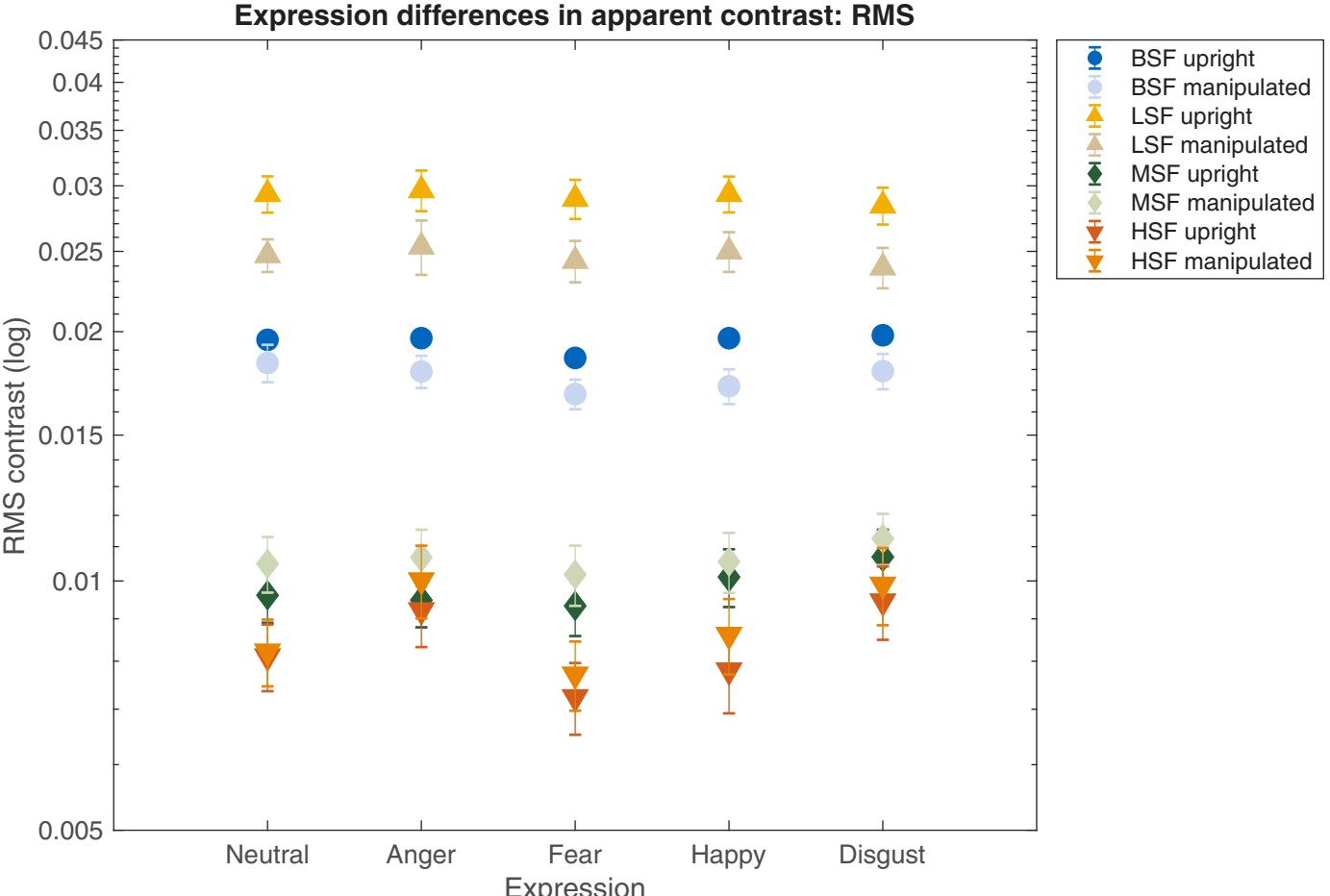

**Fig 4. Apparent contrast: RMS.** RMS contrast settings for images of facial expressions when face images are perceptually matched to a reference stimulus whose contrast is fixed at 10% Michelson contrast. Lower RMS settings denote less physical contrast required for perceptual matching, thus implying relatively better salience, and therefore higher apparent contrast. Apparent contrast is shown for broadband (blue, circle), low frequency (yellow, triangle), mid-range (green, diamond), and high frequency (orange, inverted triangle) face images. Error bars represent associated SEMs.

**Summary.** Relative to other expressions, fearful faces tend to be lower in RMS contrast as a result of a reduction in contrast primarily at high spatial frequencies. At first glance, this runs counter to the notion that the fear bias represents effects of low-level visual properties, such as contrast, on images' salience. However, it should be noted that the high spatial frequency range at which contrast is reduced contributes relatively little to perceived contrast, so in fact is unlikely to influence image salience. An important methodological point that follows from this, however, is that contrast normalisation will tend to artificially boost the effective contrast of fearful faces. Specifically, the contrast of low and midrange frequencies, which are known to drive the detection of images, and their apparent contrast, is boosted in order to compensate for a reduction in contrast at the relatively unimportant high spatial frequencies. The contrast normalisation typical in other studies is therefore expected to artificially inflate any fear biases operating in natural viewing.

## Experiment 2: Contrast matching

Michelson and RMS contrast were calculated for each face at the point when observers reported that its contrast perceptually matched that of a 10% Michelson contrast reference.

RMS and Michelson contrast settings for perceptually matched expressions are shown in Figs 4 and 5, respectively. Lower settings denote that faces required *less* physical contrast for observers to perceptually match them to a reference face, such that the target stimuli appeared relatively high in contrast when physically matched. Settings were made for upright faces, and for those which had been inverted, and had their luminance polarity reversed. At each frequency condition, a two-way repeated measures ANOVA explored effects of expression (neutral, anger, fear, happy, disgust) and manipulation (normal, manipulated). Sidak-corrected comparisons explored expression-related effects between fear other faces.

**Broadband faces.** Data are summarised in S3 Table. A two-way repeated measures ANOVA showed significant effects of expression and manipulation ($F(4, 72) = 7.72$, $p < .001$, $\eta p^2$ .30; $F(1, 18) = 9.89$, $p = .006$, $\eta p^2$ .35, respectively) on set RMS contrast, but no significant interaction. Manipulated faces (inverted with retained luminance polarity) were set to lower contrasts than unmanipulated faces; a finding consistent with previous research [25]. Statistically significant sidak-corrected comparisons showed that in order to perceptually match a reference face, upright fear expressions require 5.41, 5.95 and 7.03% *less* RMS contrast than neutral, happy and disgust faces (respectively). In the manipulated condition, control fear expressions require 6.55 and 8.86% *less* RMS contrast than angry and neutral controls (respectively). No other significant differences were observed. Data are illustrated in Fig 4.

For Michelson contrast settings, significant effects of expression and manipulation were found ($F(4, 72) = 3.56$, $p = .01$, $\eta p^2$.16; $F(1, 18) = 10.39$, $p = .005$, $\eta p^2$.36, respectively), but no significant interaction. Manipulated faces required less Michelson contrast when perceptually matched compared to their normally presented counterparts. Sidak-corrected comparisons showed no significant differences between contrast settings between fear and other faces. Data are illustrated in Fig 5.

**Low frequency filtered faces.** Data are summarised in S4 Table. For RMS contrast settings, manipulated face images required less RMS contrast compared to normal faces in order to be perceptually matched ($F(1, 18) = 41.82$, $p < .001$, $\eta p^2$ .69), but there was no effect of expression, and no interaction. Data are illustrated in Fig 4.

For Michelson contrast settings, significant effects of expression and manipulation were observed ($F(4, 72) = 2.58$, $p = .04$, $\eta p^2$ .12; $F(1, 18) = 42.73$, $p < .001$, $\eta p^2$ .70, respectively), but no significant interaction. Manipulated faces required less Michelson contrast than normally presented faces in order to be perceptually matched. Sidak-corrected comparisons did not reveal any significant differences in set contrast between fear expressions their emotion counterparts. Data are illustrated in Fig 5.

**Mid-range frequency filtered faces.** Data are summarised in S5 Table. For RMS contrast settings, significant effects of expression and manipulation were observed ($F(4, 72) = 11.26$, $p < .001$, $\eta p^2$ .38; $F(1, 18) = 6.34$, $p = .02$, $\eta p^2$ .26, respectively), but no interaction. In this case, manipulated face images required *more* RMS contrast compared to normally presented counterparts in order to perceptually match the reference. Statistically significant sidak-corrected comparisons showed that in order to perceptually match a reference face, upright mid-range frequency fear expressions require 14.6% *less* RMS contrast than disgust faces. In the manipulated condition, control mid-range frequency fear expressions required 10.51% *less* RMS contrats than disgust controls. No other significant differences were found. Data are illustrated in Fig 4.

For Michelson contrast settings, a significant effect of expression and manipulation were observed ($F(4, 72) = 13.91$, $p < .001$, $\eta p^2$ .43; $F(1, 18) = 6.46$, $p = .02$, $\eta p^2$ .26, respectively), and a significant interaction ($F(4, 72) = 2.55$, $p = .04$, $\eta p^2$ .12). Mid-range manipulated face images were again perceived as lower in contrast compared to normally-presented counterparts. Statistically significant sidak-corrected comparisons showed that in order to perceptually match a

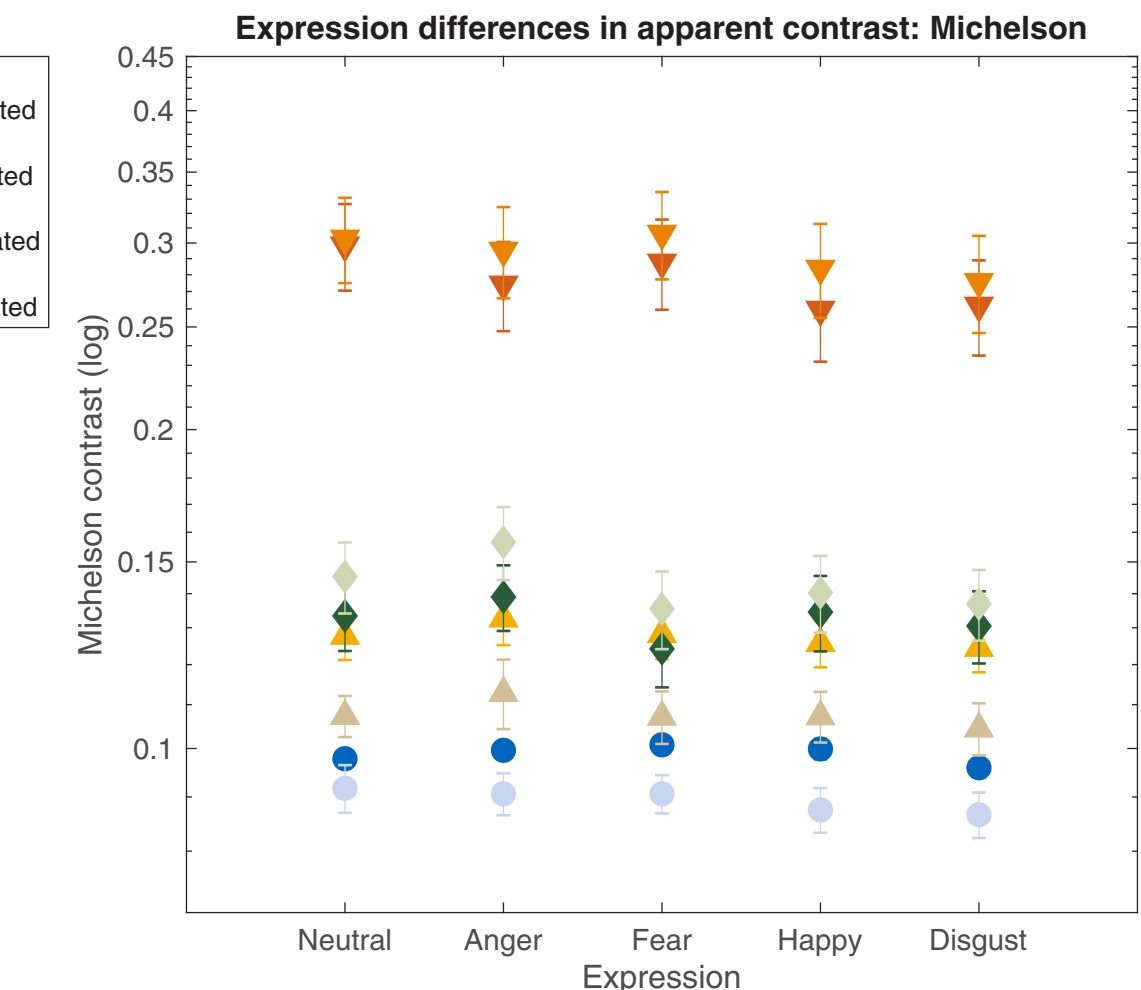

**Fig 5. Apparent contrast: Michelson.** Michelson contrast settings for images of facial expressions when face images are perceptually matched to a reference stimulus whose contrast is fixed at 10% Michelson contrast. Lower RMS settings denote less physical contrast required for perceptual matching, thus implying relatively better salience, and therefore higher apparent contrast. Apparent contrast is shown for broadband (blue, circle), low frequency (yellow, triangle), mid-range (green, diamond), and high frequency (orange, inverted triangle) face images. Error bars represent associated SEMs.

reference face, upright mid-range frequency fear expressions require 7.41 and 11.92% *less* Michelson contrast than neutral and angry faces (respectively). In the manipulated condition, control fear expressions require 6.55 and 8.86% *less* Michelson contrast than angry and neutral controls (respectively). In the manipulated condition, mid-range control fear expressions required 15.57 and 7.23% *less* Michelson contrast than angry and neutral controls (respectively). No other significant differences were observed. Data are illustrated in Fig 5.

**High frequency filtered faces.** Data are summarised in S6 Table. For RMS contrast settings, significant effects of expression and manipulation were observed ($F(4, 72) = 26.04$, $p <$ .001, $\eta p^2$ .59; $F(1, 18) = 7.33$, $p = .01$, $\eta p^2$ .28, respectively), but no significant interaction. Manipulated faces required more RMS contrast than normally-presented counterparts perceptually match the reference. Statistically significant sidak-corrected comparisons showed that in order to perceptually match a reference face, upright high frequency fear expressions require 12.5, 27.3 and 20.52% *less* RMS contrast than neutral, angry and disgust faces (respectively). In the manipulated condition, control high frequency fear expressions require 6.49, 29.96, 11.67

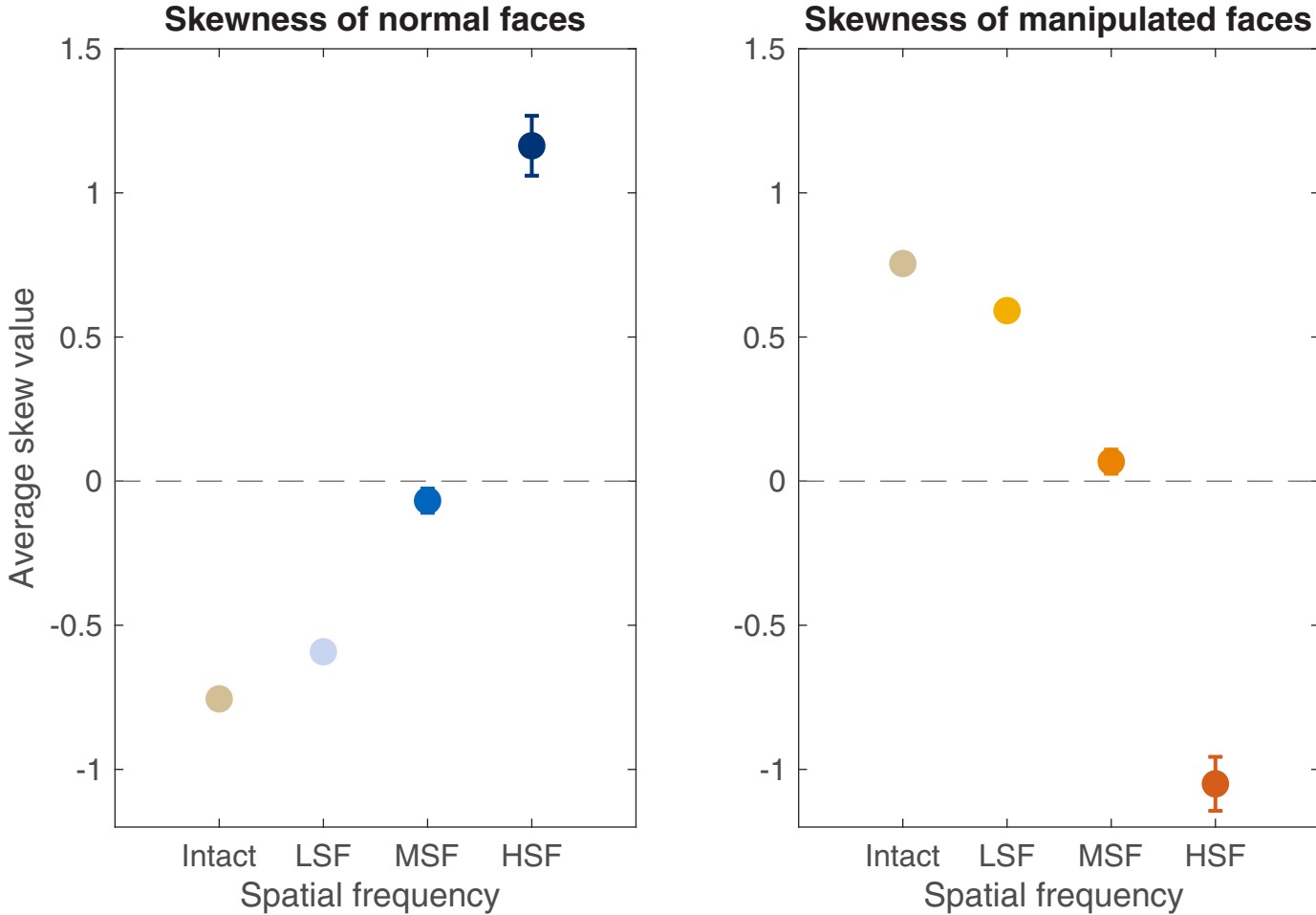

**Fig 6. Skewness.** Average skewness values for broadband, low, mid-range, and high frequency filtered versions of faces. Skew values are pooled across expression within each frequency category. Error bars represent associated SEMs. Note: some bars are smaller than marker sizes.

and 28.4% *less* RMS contrast than neutral, angry, happy and disgust controls (respectively). Data are illustrated in Fig 4.

For Michelson contrast settings, significant effects of expression and manipulation were observed ($F(4, 72) = 10.49$, $p < .001$, $\eta p^2$ .36; $F(1, 18) = 8.05$, $p = .01$, $\eta p^2$ .30, respectively), but no interaction. Manipulated faces still required more Michelson contrast than normally-presented counterparts to perceptually match the reference. Statistically significant sidak-corrected comparisons showed that -unlike effects observed for all other frequency conditions- in order to match a reference face, upright high frequency fear expressions require 9.74 and 8.94% *more* Michelson contrast than happy and disgust faces (respectively). In the manipulated condition, control high frequency fear expressions require 7.32 and 9.96% *more* Michelson contrast than happy and disgust controls (respectively). No other significant differences were observed. Data are illustrated in Fig 5.

**Summary.** The physical contrast required for faces to match a reference stimulus varies between expressions, spatial frequency filtering, contrast polarity and contrast metric. Broadband expressions differ in perceived contrast, but only when RMS contrast is the metric used; a fear advantage for these faces becomes more apparent as their spatial frequency content

increases. Differences in perceived contrast between expressions appears to be less pronounced when Michelson contrast is the metric used, and if the spatial frequency content is lower.

## Experiment 3: Post-hoc analysis of image skewness

Data from Experiment 2 showed that manipulated faces, compared to natural (upright, retained luminance polarity) faces, require less physical contrast to be perceptually matched, suggesting that they are perceived as higher in contrast. This was true for broadband and low frequency stimuli, the two conditions containing low-frequency information. At mid-range and high frequency conditions, the increased contrast for manipulated faces was abolished. One explanation of these results is that the salience associated with manipulated faces is a consequence of luminance polarity reversal. Haun and Peli [25] propose that darker image regions inform judgements of apparent contrast more than lighter regions of the same local contrast. The following post-hoc analysis of the skewness of our facial stimuli seeks to confirm that the pixel intensity distribution of normal faces becomes negatively skewed under conditions of manipulation.

To assess skewness differences between normal and manipulated facial expressions, a two-way Expression (neutral, anger, fear, happy, disgust) x Manipulation (normal, manipulated) repeated measures ANOVA was performed for each spatial frequency condition. Average skewness values, pooled across facial expression, for normal and manipulated faces are displayed in Fig 6. A positive average skewness value indicates that pixel intensities are biased towards darker values of the spectrum, and a negative average skewness value indicates pixel intensities biased towards the brightest points of the spectrum.

**Broadband faces.** A significant effect of manipulation ($F(1, 15) = 857.01$, $p < .001$, $\eta p^2$ .98) and expression by manipulation interaction were observed ($F(4, 60) = 28.81$, $p < .001$, $\eta p^2$ .66). Luminance distribution of manipulated faces was more positively skewed compared to normal faces. This finding confirms that broadband manipulated faces contain a higher proportion of dark pixels, accounting for their more visible appearance. No further analyses were performed.

## Low frequency faces

A significant effect of manipulation ($F(1, 15) = 466.21$, $p < .001$, $\eta p^2$ .96) and expression by manipulation interaction were observed ($F(4, 60) = 45.85$, $p < .001$, $\eta p^2$ .75). No significant effect of expression was observed. Similarly, to broadband facial stimuli, the luminance distribution of low frequency manipulated faces is more positively skewed compared to normal face counterparts. No further analyses were performed.

**Mid-frequency faces.** No significant effects of manipulation or expression were observed. There was, however, a significant expression by manipulation interaction ($F(4, 60) = 4.27$, $p = .05$). No further analyses were performed.

**High-frequency faces.** A significant effect of manipulation ($F(1, 15) = 136.38$, $p < .001$, $\eta p^2$.90) and expression by manipulation interaction ($F(4, 60) = 34.54$, $p < .001$, $\eta p^2$ .69) were observed. There was no significant effect of expression. In contrast to effects found for both broadband and low frequency conditions, luminance distribution of high frequency manipulated faces was in fact more *negatively* skewed compared to normal face counterparts. This effect reflects the greater number of dark pixels in high frequency manipulated faces. Importantly, this shows that as the spatial frequency content of faces increases, their appearance becomes darker when they are subjected to image manipulation.

**Summary.** These findings confirm that the pixel intensity distributions are skewed negatively for natural faces and therefore positively for manipulated faces. This reflects the

comparably greater number of bright pixels in manipulated faces; a feature that is likely responsible for their lower apparent contrast than polarity reversed images. This is consistent with the proposal that darker regions of an image, including those with a negated-appearance, appear more visible and higher in perceived contrast [25]. The data clarify that advantages in apparent contrast are likely driven by the first- and second-order statistics of manipulated faces, rather than their "face" information. Of equal importance is the effects of spatial frequency filtering. Data from Experiment 2 showed that at mid-range and high-frequency conditions, manipulated faces were not perceived as higher in contrast compared to normal faces. Our subsequent analysis suggests this is probably due to the luminance distribution faces, which were negatively skewed as a result of their low frequency content only.

## Discussion

The present study extends the current understanding of the way in which facial expressions differ in terms of their physical composition, and importantly, how such differences are affected by spatial filtering techniques. Recent findings show that facial stimuli differ compared to images of natural scenes in terms of their Fourier amplitude spectra, and that differences also exist *within* face categories, where fear expressions tend to be lower in RMS contrast compared to other expressions [23, 31]. Our findings support the notion that facial expressions are associated with naturally-occurring differences, both at the physical and subjective level. This is true in terms of the physical RMS contrast and Fourier amplitude spectra (Experiment 1), and the way in which expressions differ in terms of their perceived salience, measured by apparent contrast (Experiment 2). Moreover, the present studies extend this understanding, confirming that the RMS contrast of fearful faces tends to be lower than that of other expressions, but also that this tendency for lower RMS contrast for fear expressions becomes increasingly apparent as faces are filtered to contain only high spatial frequencies. These findings have important consequences for studies of facial expressions that employ spatial filtering techniques to explore "low road" approaches to the threat bias for fear expressions. Findings show that combinations of spatial filtering and contrast normalisation will tend to alter the relative contrast of stimuli compared to that found under natural viewing, where these effects disproportionately alter the physical and perceived appearance between expressions of different emotions.

Taken together, the findings show that naturally presented fearful expressions intrinsically require *less* physical contrast in order to appear equal in terms of subjective salience. This means that physically contrast normalised fearful faces, because they are naturally lower in physical RMS contrast, receive an inadvertent boost in perceived contrast when they are subjected to contrast normalisation, particularly at the low and midrange spatial frequencies which are known to be important in the threat bias. In other words, facial stimuli normalised for contrast may be equal at the physical level, but will remain significantly different in terms of their salience.

Together, these findings have important implications for current theories of the fear expression bias, facial expression perception in general, and our understanding of the effects that tools used for stimulus standardisation have upon behavioural and perceptual performance. They highlight the importance of incorporating ecological approaches to stimulus design reflect natural viewing, particularly for studies interested in the perceptual salience of facial expressions, and for those that use spatial filtering techniques to explore the mechanisms of these effects. In terms of their application to broader aspects of psychology, our data show that in contexts where contrast normalisation is a necessary procedure, face images normalised for Michelson contrast are likely to be assured a better degree of consistency between their

physical and perceived salience. Finally, the initially lower physical contrast of fear expressions means that low-to-midrange spatial frequencies will be artificially boosted through contrast normalisation, which may be an inadvertent, artefactual contributing factor in the perceptual bias for fearful faces.

## Supporting information

**S1 Table. RMS contrast differences between fear and expression counterparts.** Sidak-corrected paired comparisons ($\alpha = 0.0127$) are performed separately for each frequency category: broadband, low-, mid-, and high-frequency versions of faces. $df = 139$ for all tests.
(DOCX)

**S2 Table. Fourier amplitude spectra.** Fourier amplitude spectrum between fear expressions and neutral, anger, happy and disgust face counterparts. Paired comparisons are Sidak-corrected paired comparisons ($\alpha = 0.0127$). $df = 139$ for all tests.
(DOCX)

**S3 Table. Apparent contrast comparisons for broadband faces.** Sidak-corrected paired comparisons ($\alpha = 0.0063$) between broadband fear and emotion counterparts. Comparisons in the first instance are calculated using RMS contrast, and Michelson contrast in the second instance. For both sets of comparisons an additional 4 tests were included to identify whether differences are preserved under conditions of manipulation. $df = 18$ for all comparisons.
(DOCX)

**S4 Table. Apparent contrast comparisons for LSF faces.** Sidak-corrected paired comparisons ($\alpha = 0.0063$) between low-frequency (LSF) fear expressions and emotion counterparts. Comparisons are calculated using Michelson contrast only; no significant expression effect was observed for the same faces when RMS contrast was the metric. $df = 18$ for all comparisons.
(DOCX)

**S5 Table. Apparent contrast for MSF faces.** Sidak-corrected paired comparisons ($\alpha = 0.0063$) between mid-range frequency (MSF) fear expressions and emotion counterparts. Comparisons in the first instance are calculated using RMS contrast, and Michelson contrast in the second instance. For both sets of comparisons an additional 4 tests were included to identify whether differences are preserved under conditions of manipulation. $df = 18$ for all comparisons.
(DOCX)

**S6 Table. Apparent contrast for HSF faces.** Sidak-corrected paired comparisons ($\alpha = 0.0063$) between high frequency (HSF) fear expressions and emotion counterparts. Comparisons in the first instance are calculated using RMS contrast, and Michelson contrast in the second instance. For both sets of comparisons an additional 4 tests were included to identify whether differences are preserved under conditions of manipulation. $df = 18$ for all comparisons.
(DOCX)

## Author Contributions

**Conceptualization:** Abigail L. M. Webb.

**Data curation:** Abigail L. M. Webb.

**Formal analysis:** Abigail L. M. Webb.

**Funding acquisition:** Abigail L. M. Webb.

**Investigation:** Abigail L. M. Webb.

**Methodology:** Abigail L. M. Webb.

**Project administration:** Abigail L. M. Webb.

**Resources:** Abigail L. M. Webb.

**Software:** Abigail L. M. Webb, Paul B. Hibbard.

**Supervision:** Abigail L. M. Webb, Paul B. Hibbard, Rick O'Gorman.

**Validation:** Abigail L. M. Webb, Paul B. Hibbard.

**Visualization:** Abigail L. M. Webb.

**Writing – original draft:** Abigail L. M. Webb, Rick O'Gorman.

**Writing – review & editing:** Abigail L. M. Webb, Paul B. Hibbard.

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
