## [Decision Letter · Decision Letter 0]

24 Apr 2020

PONE-D-20-02214

Contrast normalisation masks natural expression-related differences and artificially enhances the perceived salience of fear expressions.

PLOS ONE

Dear Dr Webb,

Thank you for submitting your manuscript to PLOS ONE. After careful consideration, we feel that it has merit but does not fully meet PLOS ONE’s publication criteria as it currently stands. Therefore, we invite you to submit a revised version of the manuscript that addresses the points raised during the review process.

We would appreciate receiving your revised manuscript by Jun 08 2020 11:59PM. To enhance the reproducibility of your results, we recommend that if applicable you deposit your laboratory protocols in protocols.io, where a protocol can be assigned its own identifier (DOI) such that it can be cited independently in the future. For instructions see: http://journals.plos.org/plosone/s/submission-guidelines#loc-laboratory-protocols

We look forward to receiving your revised manuscript.

Kind regards,

Zezhi Li, Ph.D., M.D.

Academic Editor

PLOS ONE

Journal Requirements:

2. Please include captions for your Supporting Information files at the end of your manuscript, and update any in-text citations to match accordingly. Please see our Supporting Information guidelines for more information: http://journals.plos.org/plosone/s/supporting-information

Reviewers' comments:

Reviewer's Responses to Questions

**Comments to the Author**

1. Is the manuscript technically sound, and do the data support the conclusions?

Reviewer #1: Yes

Reviewer #2: Partly

2. Has the statistical analysis been performed appropriately and rigorously? 

Reviewer #1: Yes

Reviewer #2: Yes

3. Have the authors made all data underlying the findings in their manuscript fully available?

Reviewer #1: No

Reviewer #2: Yes

4. Is the manuscript presented in an intelligible fashion and written in standard English?

Reviewer #1: Yes

Reviewer #2: Yes

5. Review Comments to the Author

Reviewer #1: The manuscript (PONE-D-20-02214) describes a thoughtful study on an interesting topic; in addition, the manuscript is very well written.

The overall logic and methodology used appear to be sound.

The authors indicated that the participants all "had normal or corrected to normal vision". Did the authors actually measure their participants' visual acuity, or did they just assume that it was "normal"? If the authors measured acuity, please report average acuity for the group (and range of measured acuities).

Procedure, lines 164-166. The authors say that the first session collected data from 19 participants. They then say that "data collection was added for an additional 19 participants". Does this mean that there were a total of 38 participants? If so, this conflicts with the previous specification (on line 138) that "nineteen individuals took part in the study". These statements are confusing and apparently contradictory; please clarify.

The manuscript says that the experimental data are "under embargo" and includes a "private link". My understanding is that the published article in PLOS ONE should provide readers with access to all relevant data. While access to the data is clearly provided to the reviewers, I am expecting the authors to make the data available publicly as well. Please confirm.

With regards to Figure 1 and the associated pairwise comparisons, I will trust the authors' reports of significant differences between the contrasts associated with fear and the other emotions, but the differences look so small. What are the numerical differences? Are these differences in contrast (while statistically significant), large enough to be practically important? Can the authors plot these results in a way that the obtained differences are more perceptually visible to readers? What about plots of contrast DIFFERENCE between fear and the other emotions (instead of absolute magnitudes, as shown in Figure 1). As things stand now, the reported differences don't look compelling (in both Figures 1 and 2). It is really hard to see meaningful differences in the slopes of the curves (a significant difference in slope was reported in the manuscript between fear and the other expressions) shown in Figure 2.

With regards to the matching task, from the figures provided, it is not readily evident from the plots that fear is all that different from the other expressions, despite the reported statistical outcomes. Are the small statistical differences found in this study between fear and the other expressions truly of a magnitude that creates practical and meaningful effects? I trust that this is so, but these are my thoughts. The authors' conclusions sound important, but the differences between fear and the other expressions usually appear so small in the manner that they are plotted now.

The error bars shown in Figure 1 are not described in the figure caption (standard deviations?, standard errors?)

The error bars shown in Figure 5 are not described in the figure caption (standard deviations?)

Reviewer #2: Dear author,

The manuscript was very interesting to read and I have a series of comments that I have elaborated with reference to the lines of the manuscript. As you will see, most of my comments relate to clarifications. Overall I think that, despite the fact that data are clear, and results are clear, the structure of the methods needs clarity and the discussion could be ameliorated by not leaving out some of the results' elements that I think are crucial.

L.112. It is not fully clear how or in which way the present study extends previous findings. Authors should be more explicit. For instance, I would invite the authors to openly state what is the specific advancement of this work compared to previous research.

L.121 and L.136 :

It would be clearer to call this subsection "experiment 1" and then "Experiment 2" such as to make it consistent with what is said in L.112 and L.114. And present each section as a separate experiment.

L133. Can the authors please make the MATLAB code used for both features available? otherwise specify specifically the steps taken to measure both features.

L.159. I beliebe that the correct 1997 reference by E.Peli (#26) to which the authors refer to is "Pelli D. G. (1997). The VideoToolbox software for visual psychophysics: Transforming numbers into movies. Spatial Vision, 10, 437–442." And not the one titled "In search of a contrast metric...". Is this possible?

L.161 Could the authors please add visuals to illustrate the procedure timelines?

L164. Did the trials include both instances of faces (normal and manipulated?). If so, please specify here (again) too.

L165.Can the authors please justify in the text why they tested the MSF faces separately?.

L.174. It is a bit unclear why it is stated in this title that the results are those between broadband AND spacially filtered expressions but later (L178) it is specified that the ANOVA is for each spacial frequency separately. This should be consistent.

L.177. Please check again the bar graphs of Figure 1. They do not seem correct. They all look flat. Overall, I would advice that the authors use the SEM in their error bars everywhere.

L177. Could the authors please specify in the text on what are the ANOVAs being performed?. (e.g., "Repeated measures ANOVA was performed on..... for each spatial frequency". It will help the reader.

L182. Please check the reporting style. equality symbols are constantly missing throughout the manuscript.

L.370 The authors first talk about two experiments..which actually point out to two stages of their procedure. They later present results in one major section, instead of making explicit that there are two results section and finally introduce an "experiment 3", which is actually the results of post-hoc analyses which are actually subsequent analyses added to further understand the results of the "experiment 2"...

I think that the authors must clarify the parts in the manuscripts' methods and results section and be consistent with what they present. Perhaps a structure in "Experiment 1" and then "Experiment 2" totally separate could bring clarity.

L.384. I believe that the discussion section is lacking some development relative to the emotion-emotion comparisons, either in the RMS analyses or the contrast matching. I wonder if the authors can elaborate more on these. I think this is key to understand the mechanisms by which fear may or may not have the saliency effects and crucially bring more insight into the "adaptive pressure" vs. "low-level" tenets to which they refer in the introduction. For instance, it seems that in many of their results (e.g., the pairwise comparisons) that happy facial expressions are somehow immune to contrast differences compared to fearful faces. But also the rest of the comparisons are also important to mention. This could be implemented in a paragraph in the discussion.

L.418 what do the error bars represent in Fig1?.

As you can see, overall, my comments are about clarity, mainly in the structure of presentation of methods and results. Moreover, I think the authors should elaborate a bit more on their discussion which seems slightly meagre, although they had very interesting results that are not talked about or discussed later on. I think this is important, particularly considering the importance of their findings for emotion research in general. Finally, I think they should add more visuals to the manuscript.

Thank you.

6. PLOS authors have the option to publish the peer review history of their article (what does this mean?). If published, this will include your full peer review and any attached files.

Reviewer #1: No

Reviewer #2: No

---

## [Author Response · Author response to Decision Letter 0]

7 May 2020

Please see the document attached - Response to Reviewers'- for detailed responses to each suggestion proposed by both Reviewers.

---

## [Decision Letter · Decision Letter 1]

28 May 2020

Contrast normalisation masks natural expression-related differences and artificially enhances the perceived salience of fear expressions.

PONE-D-20-02214R1

Dear Dr. Webb,

We are pleased to inform you that your manuscript has been judged scientifically suitable for publication and will be formally accepted for publication once it complies with all outstanding technical requirements.

With kind regards,

Zezhi Li, Ph.D., M.D.

Academic Editor

PLOS ONE

Additional Editor Comments (optional):

Reviewers' comments:

Reviewer's Responses to Questions

**Comments to the Author**

1. If the authors have adequately addressed your comments raised in a previous round of review and you feel that this manuscript is now acceptable for publication, you may indicate that here to bypass the “Comments to the Author” section, enter your conflict of interest statement in the “Confidential to Editor” section, and submit your "Accept" recommendation.

Reviewer #1: All comments have been addressed

Reviewer #2: All comments have been addressed

2. Is the manuscript technically sound, and do the data support the conclusions?

Reviewer #1: (No Response)

Reviewer #2: Yes

3. Has the statistical analysis been performed appropriately and rigorously? 

Reviewer #1: (No Response)

Reviewer #2: Yes

4. Have the authors made all data underlying the findings in their manuscript fully available?

Reviewer #1: (No Response)

Reviewer #2: Yes

5. Is the manuscript presented in an intelligible fashion and written in standard English?

Reviewer #1: (No Response)

Reviewer #2: Yes

6. Review Comments to the Author

Reviewer #1: (No Response)

Reviewer #2: (No Response)

7. PLOS authors have the option to publish the peer review history of their article (what does this mean?). If published, this will include your full peer review and any attached files.

Reviewer #1: No

Reviewer #2: No

---

## [Editor Report · Acceptance letter]

3 Jun 2020

PONE-D-20-02214R1 

Contrast normalisation masks natural expression-related differences and artificially enhances the perceived salience of fear expressions. 

Dear Dr. Webb:

I'm pleased to inform you that your manuscript has been deemed suitable for publication in PLOS ONE. Congratulations! Your manuscript is now with our production department. 

Kind regards, 

on behalf of

Dr. Zezhi Li 

Academic Editor

PLOS ONE